# Small Language Models Fine-tuned to Coordinate Larger Language Models improve Complex Reasoning

**Gurusha Juneja**[*]
IIT Delhi, India

**Subhabrata Dutta**[*]
IIT Delhi, India

**Soumen Chakrabarti**
IIT Bombay, India

**Sunny Manchhanda**
DYSL-AI, India

**Tanmoy Chakraborty**[†]
IIT Delhi, India

## Abstract

Large Language Models (LLMs) prompted to generate chain-of-thought (CoT) exhibit impressive reasoning capabilities. Recent attempts at prompt decomposition toward solving complex, multi-step reasoning problems depend on the ability of the LLM to simultaneously decompose and solve the problem. A significant disadvantage is that foundational LLMs are typically not available for fine-tuning, making adaptation computationally prohibitive. We believe (and demonstrate) that problem decomposition and solution generation are distinct capabilites, better addressed in separate modules, than by one monolithic LLM. We introduce DaSLaM, which uses a decomposition generator to decompose complex problems into subproblems that require fewer reasoning steps. These subproblems are answered by a solver. We use a relatively small (13B parameters) LM as the decomposition generator, which we train using policy gradient optimization to interact with a solver LM (regarded as blackbox) and guide it through subproblems, thereby rendering our method solver-agnostic. Evaluation on multiple different reasoning datasets reveal that with our method, a 175 billion parameter LM (text-davinci-003) can produce competitive or even better performance, compared to its orders-of-magnitude larger successor, GPT-4. Additionally, we show that DaSLaM is not limited by the solver's capabilities as a function of scale; e.g., solver LMs with diverse sizes give significant performance improvement with our solver-agnostic decomposition technique. Exhaustive ablation studies evince the superiority of our modular finetuning technique over exorbitantly large decomposer LLMs, based on prompting alone.

## 1 Introduction

In recent years, an astounding variety of text and NLP tasks have been accomplished by language models (LMs) (Devlin et al., 2019) — in essence, fitting continuous feature vectors to tokens and modeling smooth conditional distributions over thousands of token positions with multi-task objectives. The next generation of large LMs (LLMs) such as T5, GPT4 and Bard (Raffel et al., 2020; OpenAI, 2023) developed protean capabilities, extending to mathematical and logical ability, based on prompting and in-context learning. Chain-of-thought (CoT) prompting has been a key enabler (Wei et al., 2022; Feng et al., 2023). LLMs can solve middle-school word problems and equations reasonably well. It has also acquired the ability to invoke specialized external tools such as Wolfram Alpha (Wolfram, 2023; Schick et al., 2023).

Recent advances in LLMs have arisen largely from cleverly-engineered, ever-growing training data, rather than any significant change in network structure, which remains relatively regular, but with rapidly increasing network size and number of parameters. One outcome of such giant monolithic LLMs is unsustainable levels of hardware and energy (Dhar, 2020) to train them. Meanwhile, neurologists and brain scientists have known, via fMRI scans, *inter alia*, that cerebral functions are specialized and spatially localized (Fedorenko and Varley, 2016; Mahowald et al., 2023).

Many recent complex reasoning challenges thrown at LLMs have a two-level character – the input task needs to be decomposed into subtasks, then the subtasks need to be solved, and finally, subtask solutions have to be consolidated and combined to solve the input task. Existing approaches use the same LLM to both decompose and solve the task, sometimes in tangled and uninterpretable ways. Because the sharing of an LLM across these functions cannot be closely controlled, very large models are needed for this double ability (decompose and solve) to emerge.

Staying entirely inside the LLM regime, and avoiding the possibility of specialized tools, we

---

[*]Equal contribution as first author.
[†]Correspondence: tanchak@iitd.ac.in

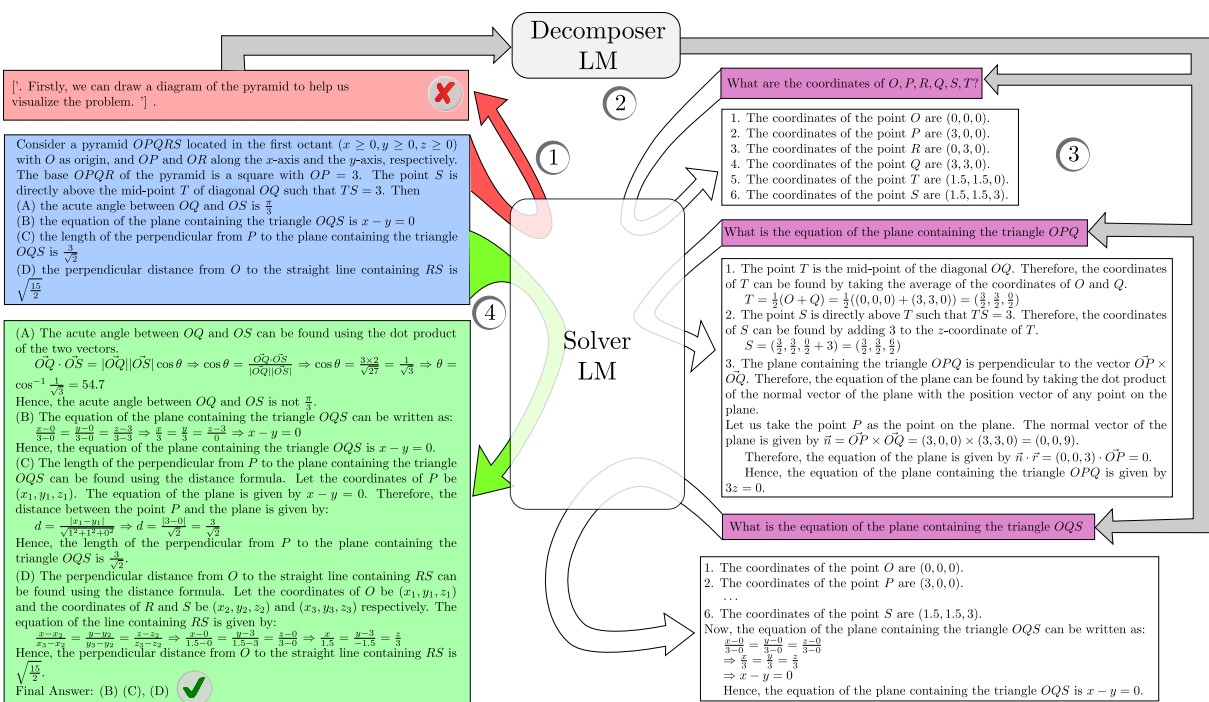

Figure 1: **Working example of DaSLaM** on a mathematical reasoning question from the JEEBench dataset (Arora et al., 2023). In this example, the solver LM is text-davinci-003. In step ①, the solver is prompted to answer the question (blue textbox) and it fails to answer correctly (red textbox). A problem decomposing LM generates subproblems (violet textboxes) conditioned on the original question and the initial response of the solver in step ②. In step ③, the solver answers these subproblems iteratively and appends to the prompt. Finally, the original problem is appended to the prompt in step ④, and the solver answers it correctly (green textbox).

ask a simple question – *is it possible to offload the ability of problem decomposition to a dedicated, relatively smaller scale model*, which is specialized and can act in synergy with any solver model of choice? To incorporate flexibility and better generalization, an immediate requirement of such a setup would be to enable a model-agnostic communication between the decomposer and the solver.

**Our contributions.** To study this research question, we develop DaSLaM (Decomposition And Solution LAnguage Models), in which we separate the decomposer from the solver, as shown in Figure 1. The solver LM can be conventionally trained or fine-tuned. In the illustration, when it answers a question incorrectly, the decomposer LM takes over to produce sub-questions. The partial solutions are appended and resubmitted to the solver LM, which solves the question correctly. The decomposer LM regards the solver as a black box, and uses reinforcement learning (RL) to become a specialized expert at decomposition, informed by the solver's mistakes.

Extensive experiments with three reasoning datasets (MATH, AQuA, and JEEBench) show that the proposed specialization improves the perfor-

mance of OpenAI GPT-3 text-davinci-003 to outperform GPT-3.5 and even begins to compete with GPT-4, outperforming other similar methods. DaSLaM boosts text-davinci-003 from an exact match accuracy of $41.6$ to $54.5$ in zero-shot regime, which is 3.9 points higher than few-shot GPT-4. Similarly, on Physics problems from JEEBench dataset, DaSLaM-augmented text-davinci-003 scores only $0.58$ points short of GPT-4 while outperforming GPT-3.5. The decomposer LM in DaSLaM reduces decomposition errors, and generalizes well across diverse small-scale LMs. It is also more robust in the face of difficult datasets, where the solver gives near-random performance.

These results support our founding hypothesis that heterogeneous functional specialization improves model efficiency and robustness of LLMs. A crucial findings from our experiments is that finetuning the decomposer is much more powerful choice than finetuning the solver. Moreover, a finetuned decomposer is largely superior compared to an orders of magnitude larger LLM prompted to act as a decomposer. Given the prohibitive cost of finetuning LLMs like GPT 3, 3.5, or 4, we hope this method would provide us a promising direction to-

wards future development of task-expert models.[1]

## 2 Related Work

Eliciting superior reasoning abilities in LM through specially designed prompts has found its popularity through CoT prompting (Wei et al., 2022) – asking the LM to explain the reasoning steps improves overall performance. Decomposing a complex reasoning requirement into multiple, simple steps results in superior reasoning capabilities across modalities other than free-form natural language text as well, e.g., reasoning over tabular data (Ye et al., 2023), visual question answering (Lu et al., 2022), etc. These methods generally solicit a single run of LM inference with no intermediate prompting interactions. Consequently, the LM often misses key reasoning steps or hallucinates irrelevant ones.

On the other hand, a prototypical prompter, sequentially interacting with the LM, has shown impressive performance. Progressive Hint Prompting (Zheng et al., 2023) uses such a setting; first, the LM is asked to provide a base answer. The prompt then uses the answer as a hint to the LM that progressively guides it to the final answer. Zhou et al. (2023) followed a similar direction by breaking down the problem itself. Their method, Least-to-most prompting, asks the LM to generate simpler, related problems from a complex problem. The final solution to the original question is generated by the LM conditioned upon the solution of the subproblems. A major bottleneck then becomes the solver's ability to identify the critical subproblems. Decomposing a complex task and then solving each task via multiple LLMs with their own in-context examples have been attempted as well (Dua et al., 2022; Khot et al., 2023). Recently, Shridhar et al. (2022) explored subquestion generation from complex questions as means of distilling reasoning abilities from larger LMs to smaller ones.

Our proposed method, DaSLaM makes a departure from these mentioned approaches in three particular features: (i) we seek to separate out the decomposer from the solver to get rid of the solver's limitations affecting decomposition, (ii) the decomposer acts as a plug-and-play module that can generalize to any solver, and (iii) the decomposition actuates with complete knowledge of the solver's actions.

---

[1]The codebase is given at: https://github.com/LCS2-IIITD/DaSLaM

## 3 A General Overview of DaSLaM

Given an LM $\theta$ and a question $Q$ as a sequence of tokens, a standard zero-shot prompting can be described as,

$$\hat{A} = \arg\max_{A \in \mathcal{A}} p_\theta(A|Q)$$

where $\hat{A}$ is the inferred answer, and $\mathcal{A}$ is the set of possible answers (e.g., numerical values, multiple-choice options, True/False, etc.). With a CoT prompting, the LM generates a sequence of tokens explaining the steps $S$ to reach the answer given the question. The modified process can be described as,

$$\hat{A} = \arg\max_{A \in \mathcal{A}} \left[ p_\theta(A|S, Q) \arg\max_{S} p_\theta(S|Q) \right] \quad (1)$$

When answering $Q$ requires multistep reasoning, one can conceptualize $S$ as a sequence of smaller steps $\{S'_1, S'_2, \cdots, S'_n\}$ such that the LM iteratively answers a sequence of subproblems $\{Q'_1, \cdots, Q'_n\}$ to finally reach the desired answer to the original question. Eq. 1 can then be rewritten as,

$$\hat{A} = \arg\max_{A'_n \in \mathcal{A}} \prod_i p_\theta(A'_i|S'_i, Q'_i) \arg\max_{S'_i} p_\theta(S'_i|Q'_i) \quad (2)$$

where $A'_i$ is the answer to the subproblem $Q'_i$. In the usual regime of CoT, the subproblems $Q'_i$ are implicit; the LM discovers them on its own and generates the reasoning steps and the answers. Due to the repeated multiplication in Eq. 2, any error in the initial stages quickly propagates along the chain of steps.

In DaSLaM, we seek to alleviate this problem by offloading the task of inferring the subproblems $\{Q'_i\}$ to a decomposer LM $\phi$. For a more guided problem decomposition, DaSLaM uses the answer and the steps generated by the solver LM, $\theta$ in the naive CoT regime as described in Eq. 1 to generate a set of subproblems $\{\hat{Q}_i\}_{i=1,\dots,n}$ as follows:

$$\hat{Q}_i = \arg\max_{Q'_i} p_\phi(Q'_i|\{\hat{Q}_j : j \in [1, i-1]\},$$
$$Q, \hat{A}_0, S_0), \quad (3)$$

for $i \in [1, n]$, where $\hat{A}_0$ and $S_0$ are the initial answer and reasoning steps, respectively generated by $\theta$. The solver LM $\theta$ then solves the subproblem set one-by-one similar to Eq. 2. However, instead of seeking to generate the final answer as a response to the last subproblem $\hat{Q}_n$, we append the original question at the end and let $\theta$ answer it directly given the context generated by the subproblems, their answers, and the corresponding CoTs. The four stages of workflow with DaSLaM, as described

in Figure 1, can then be summarized as follows:

$$\hat{A}_0 = \arg\max_{A \in \mathcal{A}} \left[ p_\theta(A|S_0, Q) \arg\max_{S_0} p_\theta(S_0|Q) \right]$$

$$\hat{Q}_i = \arg\max_{Q'_i} p_\phi(Q'_i|\{\hat{Q}_j\}_{j \in \{1, i-1\}}, Q, \hat{A}_0, S_0)$$

$$\hat{A}_i = \arg\max_{\hat{A}_i \in \mathcal{A}} \left[ p_\theta(A_i|S_i, \hat{Q}_i) \arg\max_{S_i} p_\theta(S_i|\hat{Q}_i) \right]$$

$$\hat{A} = \arg\max_{A \in \mathcal{A}} \Big[ p_\theta(A|S, Q)$$

$$\arg\max_{S} p_\theta(S|\{\hat{A}_i, S_i, \hat{Q}_i\}_{i \in \{1, N\}}, Q) \Big] \tag{4}$$

## 4 Learning to Decompose from Feedback

Typical LMs are not suitable to reliably perform the problem decomposition stage in Eq. 3. We seek to finetune the decomposer LM, $\phi$ for this task. Specifically, we use the LLaMA 13 billion model. Instead of full LM-tuning, we use LoRA adapters (Hu et al., 2022) for parameter-efficient training. For brevity, we will denote the adapter-augmented LM as $\phi$, although only the adapter weights are being updated while training. The whole process of teaching the LM to perform the problem decomposition comprises two successive stages: (i) subproblem construction from the original question and CoT, and (ii) policy optimization in conjunction with the solver LM. Details of the supervised data curation required for these three steps are described in Section 5. Each data sample in the supervised dataset $\mathcal{D}$ can be conceptualized as a sequence of triplets $\langle Q_{\text{gold}}, S_{\text{gold}}, A_{\text{gold}}, \mathbf{Q}'_{\text{gold}} \rangle$, where $Q_{\text{gold}}$ denotes the original question, $S_{\text{gold}}$ denotes reasoning steps in form of CoT, $A_{\text{gold}}$ denotes the answer to the question, and $\mathbf{Q}'_{\text{gold}}$ is a sequence of subproblems generated by decomposing $Q_{\text{gold}}$ (see Section 5 and Appendix A for further details on supervised data curation process).

The first stage is straightforward with the decomposer LM being finetuned using language modeling objective. In the first stage, we seek to optimize the following objective:

$$\min_\phi [- \log(p_\phi(\mathbf{Q}'_{\text{gold}}|Q, S))] \tag{5}$$

This step is somewhat similar to instruction tuning, where an LM is asked to generate some text conditioned on a context, instead of following the usual sentence completion-style behavior of the LM. Intuitively, the role of this stage is to *align* the LM to the task of the decomposer.

**Decomposition via policy network.** The previous stage inculcates the ability to decompose a problem within $\phi$. However, it is still blind to the actual errors made by the solver LM. In the next stage, we seek to make the decomposer LM work in synergy with *any* solver LM of choice. This solver agnosticism restrains $\phi$ to observe the internal representations computed by $\theta$ while solving the problem. To handle this imposed blackbox characteristics of the solver, we resort to policy gradient optimization of $\phi$ assuming $\theta$ to be part of the environment. We formalize the setup as follows.

*Elements of the environment*: The solver LM $\theta$ constitutes the core component of the environment. Given a question $Q$, the solver-generated CoT $S$ and the answer $A$ as sequences of tokens define the observation of the policy.

*State and action space*: We define the state space as $\mathcal{S}$. The initial state, $s_0 \in \mathcal{S}$, is defined by the original question $Q$ and the initial response from the solver, $S_0, A_0$, that are provided to the decomposer LM $\phi$ as input. A single timestep is defined on generation of a single token, i.e., the action $a_t$ at time $t$. Given the autoregressive nature of LM, we define the state at $t$-th timestep as $s_t = (s_{t-1}, \{a_{t-1}\})$, i.e., the token generated at $t - 1$ appended to the tokens generated till $t - 1$. Trivially, the action space is the same as $\mathcal{V}$, the vocabulary of $\phi$.

*Policy*: The decomposer LM is conceptualized as a policy network $\pi_\phi : \mathcal{S} \to \mathcal{V}$, i.e., it generates a token given the inputs and the token generated hitherto, till the end of episode $T$. Inspired by the recent success in Reinforcement Learning from Human Feedback (RLHF) with autoregressive LMs, we choose the Proximal Policy Optimization (PPO) algorithm to train the policy $\pi_\phi(a|s)$. In a typical PPO setup, we define the advantage function as follows:

$$\hat{G}_t = \sum_{i=0}^{T-t+1} (\gamma\lambda)^i [r_{t+i} + \gamma V(s_{t+i+1}) - V(s_{t+i})] \tag{6}$$

where $r_t$ is the reward at step $t$, $V(s_t) : \mathcal{S} \to \mathbb{R}$ is the value function determining the reward associated to state $s_t$, and $\gamma, \lambda$ are hyperparameters. We use the policy model augmented with a randomly initialized feedforward layer as the value function. A crucial component in on-policy learning regime is the reward function $r_t$. While the end goal of the learning is to have the solver answering correctly, the decomposer LM should receive some incremental signal aligned to its generation as well for it to stably converge to the optimal policy. With this

goal in mind, we construct the reward function as,

$$r_t = R_1 + R_2 + R_3 + R_4 + R_5 \quad (7)$$

where $R_1$ to $R_5$ are defined as follows (see Appendix B for a detailed treatment on the reward computation method). Here cos-sim represents cosine similarity. $I(x) = 1$ if $x$ is true is the indicator function.

**Entity coverage:** $R_1 = \frac{|E_{\mathbf{Q'}}|}{|E_Q|}$, where $E_{\mathbf{Q'}}$ and $E_Q$ are the sets of distinct entities in the generated subproblems and the original question, respectively.

**Consistency of answers to subproblems:**

$$R_2 = \sum_i \left( I(e_i = \hat{e}_i) + \text{cos-sim}(Q_i', A_i) \right) \quad (8)$$

where $\hat{e}_i$ is the entity whose value has been asked in the subproblem $Q_i'$, and $e_i$ is the entity answered. This reward penalizes the decomposer LM for generating questions whose answers are not consistent.

**Order of operations:** $R_3 = \frac{l}{m}$, where $l$ is the number of operations matched in order between $S$ and $S_{\text{gold}}$, and $m$ is the total number of operations in $S_{\text{gold}}$.

**CoT proximity:** To ensure that the distance of reasoning produced by the model after prompting $S$ to the gold reasoning $S_{gold}$ is less than the distance of reasoning produced without prompt $S_0$ to the gold reasoning steps $S_{gold}$, we design a reward based on the cosine similarity of each step of $S_{gold}$. We break $S$ and $S_0$ at new-line token to form reasoning steps. At each step $j$, we compute $c_{1j} = \text{cos-sim}(S^j, S_{gold}^j)$ and $c_{2j} = \text{cos-sim}(S_0^j, S_{gold}^j)$. The reward is

$$R_4 = \sum_{j=0}^m I(c_{1j} > c_{2j})c_{1j} + I(c_{2j} > c_{1j})(-1 - c_{2j}), \quad (9)$$

**Correctness of final answer:** $R_5 = I(\hat{A} = A_{\text{gold}})$. Now, we can define the PPO objective as follows:

$$\max_\phi \left( \mathbb{E}_t \left[ \frac{\pi_\phi(a_t|s_t)}{\pi_{\text{ref}}(a_t|s_t)} \hat{G}_t \right] - \beta \mathbb{E}_t \left[ K_t \right] \right) \quad (10)$$

where $\pi_{\text{ref}}$ is the reference model that is initialized with supervised finetuned $\phi$. $K_t = \text{KL}[\pi_{\text{ref}}(\cdot|s_t), \pi_\phi(\cdot|s_t)]$ is the KL-divergence between the reference model and the policy model.

The resulting decomposer LM $\phi$ optimized using the above mentioned three stages of finetuning can then be used with DaSLaM.

## 5  Experiments

**Training data curation.** The training process of DaSLaM consists of two stages as mentioned previ-

ously. In the first stage, we require the subproblems along with the reasoning steps for a given problem. We use samples from four existing datasets — MATH (Hendrycks et al., 2021), AQuA (Ling et al., 2017), GSM8K (Cobbe et al., 2021), and StrategyQA (Geva et al., 2021). Each question in these four datasets contains a question $Q_{\text{gold}}$, a step-by-step illustration of the reasoning process $S_{\text{gold}}$, and the final answer $A_{\text{gold}}$. We sample $7,000$ examples from the training splits of these datasets and employ OpenAI's text-davinci-003 model to generate the corresponding subquestions. We provide the model with one-shot example illustrating how to decompose a question into subquestions based on the reasoning. In the second stage of training, we utilize the remaining training data from MATH and AQuA datasets to conduct the policy optimization since this step does not require any supervised examples of subproblems.

**LMs used.** We use LLaMA 13 billion (Touvron et al., 2023) as the decomposer LM. For the solver LM, we primarily use text-davinci-003 (henceforth, we denote it as GPT-3.5 for brevity). We also experiment with the LLaMA 13 bilion and LLaMA 33 billion models as solvers to test the model-agnostic generalizability of DaSLaM.

**Baselines.** We compare DaSLaM with four existing methods of prompting: Chain-of-thought prompting (**CoT**) (Wei et al., 2022), Least-to-most prompting (**L2M**) (Zhou et al., 2023), Progressive Hint Prompting (**PHP**) (Zheng et al., 2023), and, Demonstrate-Search-Predict (**DSP**) (Khattab et al., 2022a). The original setting of PHP requires an 8-shot prompting; however, since all other methods including DaSLaM predict in the zero-shot setting, we use PHP in 1-shot for a fairer comparison. Additionally, we experiment with three ablation variants: **DaSLaM-NF** does not take the solver feedback into account while generating the subproblems; **Finetuned** is the solver LM (LLaMA 13B in this case, we could not finetune 33B variant due to computational constraints) finetuned without any decomposer; **GPT-3.5 decomposer** does away with the finetuned LLaMA 13B decomposer and uses pretrained GPT-3.5 as the prompted decomposer.

**Test datasets.** For evaluation purposes, we use three datasets – MATH (Hendrycks et al., 2021), AQuA (Ling et al., 2017), and JEEBench (Arora et al., 2023). For the first two datasets, only the test splits are used during evaluation since their

| Dataset | Method | | | | | | |
|---------|--------|------|------|------|-------------------|-----------|--------|
|         | CoT | L2M | PHP | DSP | GPT3.5 Decomposer | DaSLaM-NF | DaSLaM |
| PnC     | 16.4 | 16.0 | 10.2 | 16.2 | 16.0 | 20.0 | **21.4** |
| NT      | 14.4 | 11.0 | 9.8 | 20.3 | 14.2 | 18.4 | **26.1** |
| ALG     | 27.6 | 22.4 | 24.0 | 15.3 | 32.1 | 31.6 | **33.4** |
| I-ALG   | 16.4 | 16.8 | 10.0 | 17.0 | 18.4 | 20.8 | **24.8** |
| Calc.   | 14.0 | 14.58 | 14.28 | 18.8 | 12.0 | 15.1 | **18.2** |
| P-ALG   | 32.3 | 28.0 | 26.5 | 28.0 | 35.5 | 38.0 | **44.0** |
| Geom.   | 14.2 | 12.5 | 14.0 | 5.2 | **22.0** | 19.04 | 21.4 |
| AQuA    | 41.6 | 44.7 | 44.4 | 44.0 | 45.4 | 53.2 | **54.5** |

Table 1: Performance comparison on MATH and AQuA datasets using GPT-3.5 as the solver LM. See Section 5 for abbreviations.

training splits are used while finetuning the decomposer. The MATH dataset contains mathematical problems on multiple different domains. We report the results on each of them separately and use the following abbreviations – **ALG**, **I-ALG**, and **P-ALG** for Algebra, Intermediate Algebra, and Pre-Algebra, respectively; **Calc** for Calculus, **Geom** for Geometry, **PnC** for Probability and Combinatorics, **NT** for Number theory. From the JEEBench dataset, we use the problems in Physics (**Phy**) and Mathematics (**Math**). Each of these two subjects has three types of problems – single-answer multiple-choice questions (**MCQ**), numerical problems (**Num**), and multi-answer multiple-choice questions (**Multi**). For all these datasets, *we use exact match criteria to evaluate the correctness of the model-inferred answers*. Details of training and inference hyperparameters and compute resource usage are provided in Appendix C.

## 6 Experimental Results

The tasks used to evaluate the performance of DaSLaM contain questions that can be answered either of the three types – numerical, single correct answer MCQ, and multiple correct answer MCQ.

**DaSLaM is better than pure prompting** We start with DaSLaM augmented with GPT-3.5 as the solver LM on MATH and AQuA datasets (see Table 1). The improvement achieved with DaSLaM prompting compared to standard CoT is staggering across all types of problems in the MATH dataset: +11.7 on Pre-Algebra, +8.4 on Intermediate Algebra, +7.7 on Number Theory, +7.2 on Geometry, +5.0 on Probability and Combinatorics, +5.8 on Algebra, and +4.2 on Calculus. The absolute improvement is even larger on the AQuA dataset, i.e., +12.9 over CoT. It is noticeable that the effects of DaSLaM are

stronger across tasks containing algebraic reasoning (AQuA, Pre- and Intermediate-Algebra, etc.) compared to Probability and Combinatorics or Calculus, which require more implicit knowledge. The performance gain achieved via DaSLaM is significantly better compared to methods like L2M or PHP. The latter methods often fail to improve over standard CoT (e.g., on Probability and combinatorics, Number Theory, and Algebra problems, L2M shows a drop in accuracy). Even when improving over CoT, their improvement is meager compared to DaSLaM. This trend entails our earlier argument in support of offloading the problem decomposition task to a specialized LM; methods that prompt the solver LM to decompose the problem lack the expertise achieved via dedicated finetuning in DaSLaM.

**Finetuned decomposer is essential.** Despite being orders of magnitude smaller, a finetuned LLaMA 13B model delivers better performance compared to GPT-3.5 as a decomposer (DaSLaM vs. GPT-3.5 generator in Table 1 and 2). This further justifies our choice of separately finetuning the decomposer and the added flexibility that it offers. In fact, finetuning the decomposer is far effective compared to finetuning the solver (DaSLaM vs Finetuned solver in Table 2).

**Feedback from the solver is important.** In the preceding paragraph, we attributed the superiority of DaSLaM over other methods to the usage of a specialized LM for problem decomposition. However, manipulating the problem decomposition upon feedback from the solver is also an important factor here. None of the existing methods does so, and therefore, remains blind towards what reasoning (and possible errors) is followed by the solver model. This is further manifested when we

| Method | Dataset | | | | | | | |
|---|---|---|---|---|---|---|---|---|
| | PnC | NT | ALG | iALG | Geom | Cal | Palg | AQuA |
| | LLaMA 13 billion | | | | | | | |
| CoT | 2.05 | 4.0 | 3.12 | 2.4 | 3.2 | 2.08 | 5.0 | 17.7 |
| L2M | 1.66 | 3.2 | 3.33 | 2.8 | 2.0 | 3.33 | 4.54 | 16.6 |
| Finetuned | 2.8 | 3.6 | 3.57 | 3.2 | 4.1 | 3.05 | 6.04 | 19.4 |
| GPT3.5 Decomposer | 2.05 | 5.0 | 4.68 | 2.8 | 2.08 | 4.0 | 6.66 | 20.4 |
| DaSLaM-NF | 2.93 | 4.8 | 4.68 | 3.2 | 4.0 | 3.9 | 6.2 | 21.6 |
| DaSLaM | **4.0** | **5.6** | **4.70** | **3.4** | **4.3** | **4.1** | **8.33** | **22.0** |
| | LLaMA 33 billion | | | | | | | |
| CoT | 2.4 | 4.16 | 4.54 | 3.7 | 4.0 | 4.0 | 5.2 | 20.0 |
| L2M | 2.38 | 4.16 | 4.2 | 6.0 | 4.25 | 5.71 | 5.55 | 21.6 |
| DaSLaM-NF | 3.2 | 5.83 | 5.6 | 5.6 | 5.1 | 5.71 | 5.2 | 22.5 |
| DaSLaM | **4.0** | **7.36** | **9.09** | **6.02** | **5.3** | **6.03** | **8.44** | **26.8** |

Table 2: Performance on MATH and AQuA with LLaMA 13 billion and LLaMA 33 billion as solvers. PHP is not reported as one-shot PHP generated randomly with both LLaMA variants. DaSLaM provides consistent improvement across all the tasks while other baseline methods mostly fail.

| Method | Dataset | | | | | | | |
|---|---|---|---|---|---|---|---|---|
| | Phy MCQ | Math MCQ | Phy Multi | Math Multi | Phy Num | Math Num | Phy Int | Math Int |
| CoT | 33.33 | 21.9 | 6.25 | 12.0 | 3.03 | 1.69 | 12.5 | 20.0 |
| PHP | 22.22 | 17.07 | 6.25 | 7.59 | 3.03 | 1.69 | 0* | 4.0 |
| L2M | 22.22 | 21.9 | 6.25 | 12.5 | 3.03 | 3.38 | 10.0 | 20.0 |
| DaSLaM-NF | 20.8 | 31.7 | 7.5 | 10.12 | 3.03 | 3.38 | 12.5 | 16.0 |
| DaSLaM | **55.55** | **36.5** | 18.75 | 16.0 | 6.06 | 10.16 | 22.5 | **24.0** |
| GPT-4 | 55.55 | 34.14 | **27.5** | **21.5** | **15.15** | **11.8** | **25.0** | 20.0 |

Table 3: Performance comparison on the JEE benchmark dataset with GPT-3.5 as the solver LM. 0* signifies that the model was not able to answer any problem in the task correctly.

compare DaSLaM with itself without the feedback module, DaSLaM-NF. While DaSLaM-NF is able to improve upon basic CoT and other prompting methods, it falls short of a decomposer LM that has access to the initial response of the solver.

**DaSLaM generalizes to smaller solvers.** An important aspect of a prompting method is its ability to work with LMs of different scales. Despite being finetuned with GPT-3.5 responses only, DaSLaM is able to improve upon the base performance of smaller scale LLaMA models as well (see Table 2). L2M prompting generally fails with both LLaMA 13 billion and 33 billion variants. On the other hand, DaSLaM, with or without feedback, almost doubles the performance of the base CoT across multiple tasks of the MATH dataset. It shows substantial improvement on AQuA as well. The importance of feedback from the solver LM usually manifests strongly in proportion to the scale of the solver.

**DaSLaM generalizes to harder problems.** Since the decomposer LM $\phi$ is trained using a subset of the training data of MATH and AQuA, we opt for a harder (in terms of benchmark performance of different LMs) reasoning evaluation on the JEEBench dataset. Table 3 summarizes the performance of the baselines and DaSLaM with GPT-3.5 as the solver LM on Mathematics and Physics questions of the JEEBench dataset. We notice that the superiority of DaSLaM manifests even more profoundly on this task compared to the former ones. Both PHP and L2M prompting absolutely fail to improve upon

basic CoT prompting, often with a sharp fall in performance (e.g., Physics MCQ questions). On the other hand, DaSLaM boosts the LMs performance, very often over 100% relative improvement (all three types of problems in Physics and numerical problems in Mathematics). Aggregated across question types, DaSLaM boosts the performance of GPT-3.5 to 22.420 in Physics and 22.07 in Mathematics. It is noteworthy that the same LM in its base setting performs near-random, i.e., 10.4 and 10.7 in Physics and Mathematics, respectively, whereas a random selection baseline gives scores of 9.6 and 10.3, respectively (Arora et al., 2023). Furthermore, GPT-3.5 with DaSLaM outperforms a better optimized candidate of the GPT series, GPT-3.5 on both these subjects (note that Arora et al. (2023) reported 18.9 and 15.7 scores with GPT-3.5 on Physics and Mathematics, respectively). **Comparison with GPT-4.** The colossal compute used by GPT-4 makes the comparison with any of its predecessors like GPT-3.5 quite unfair. However, it is tempting to observe that DaSLaM boosts the performance of GPT-3.5 often to the level of GPT-4. For example, on arithmetic problems of the AQuA dataset, DaSLaM surprisingly outperforms both zero-shot and few-shot GPT-4 (40.6 and 50.4 respectively, compared to 54.5 with GPT-3.5 and DaSLaM). On MATH dataset, DaSLaM augmented GPT-3.5 scores an aggregate of 30.23, which is better than ChatGPT (26.4) and close to GPT-4 (35.7). On JEEBench Mathematics problems, GPT-

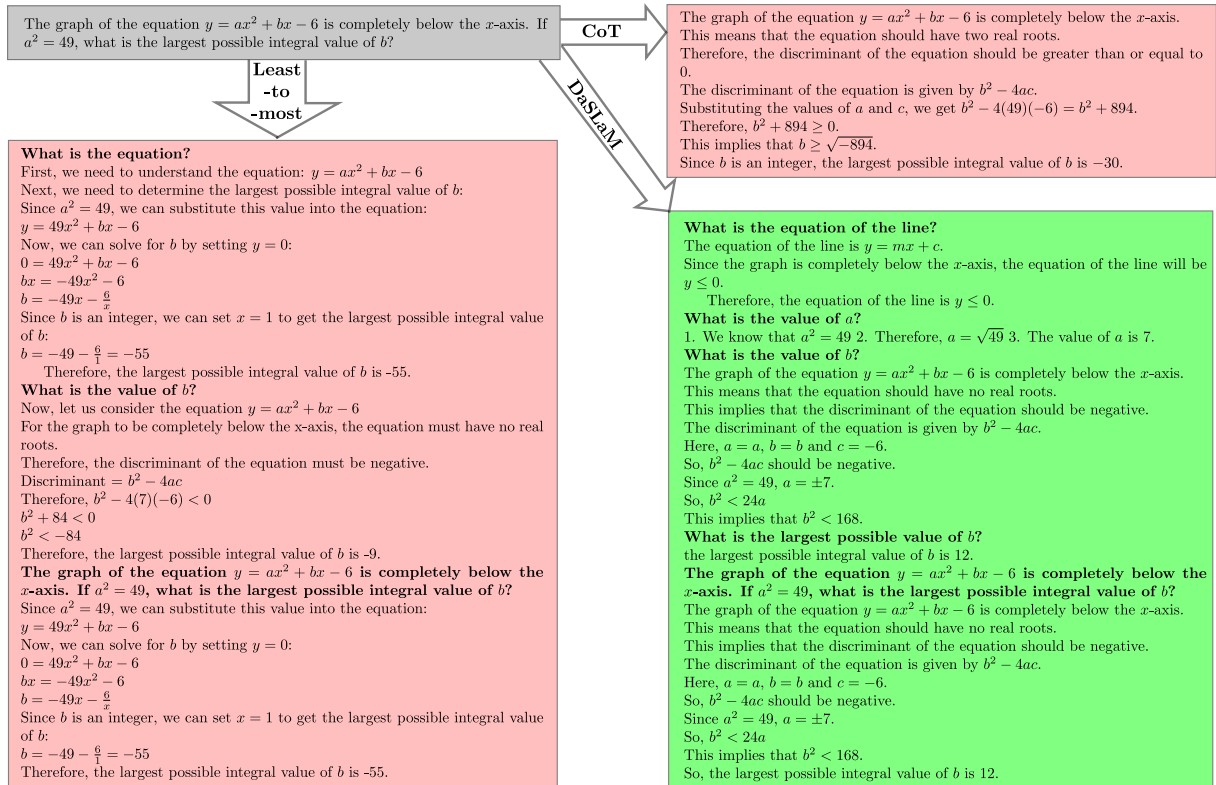

Figure 2: An example case study on a problem from the MATH dataset. GPT-3.5 is used as the solver LM with three different methods of prompting – standard CoT, Least-to-most, and `DaSLaM`. Only `DaSLaM` is able to guide the model to the correct answer.

4 comes up with an aggregate score of 23.1, which is pretty close to our 22.42. In Physics and Math MCQ questions, `DaSLaM` with GPT-3.5 outperforms GPT-4. These results definitely do not claim any assumed superiority of `DaSLaM`-boosted GPT-3.5 over GPT-4 since there are multiple other cases that state otherwise. Instead, we seek to demonstrate how much leftover potential these LMs possess that can be unleashed via our proposed method of feedback-guided automatic problem decomposition.

## 7   Case Study

To this point, we have compared the numbers produced by `DaSLaM`-boosted models across different datasets. While they provide an overall assessment, deeper analyses are needed to comprehend the actual reasoning steps adopted by these different methods. Figure 2 shows the reasoning steps generated by GPT-3.5 given an example problem from the MATH dataset with three different prompting methods – vanilla CoT, L2M, and `DaSLaM`. Note that `DaSLaM` uses the CoT output to decompose the problem.

Both CoT and L2M end up with the model an-

swering incorrectly. With CoT, the solver wrongly assumes that the given equation must have two real roots though it should not have any real roots. Also, it mistakes the value of $a^2$ as $a$. The effect is prominent in the subproblems generated by `DaSLaM` as it asks to find the value of $a$ explicitly. Furthermore, the solver LM specifically announces that $y \leq 0$ to answer the first subproblem generated by `DaSLaM`. This helps to correct the reasoning about the sign of the discriminant.

With L2M, the confusion around the value of $a$ and $a^2$ persists, as the solver LM substitutes $a$ in the given equation by 49 (which is the value of $a^2$) twice in the answering process. Although it substituted the correct value of $a$ once while answering the second question, it is not explicitly declared like in `DaSLaM`. We observe multiple similar failure cases with L2M. It is quite likely that prompting the model to generate the final answer after each subproblem accumulates the erroneous reasoning steps that the model falls prey to.

With `DaSLaM`, the reasoning steps followed by the solver remains robust throughout. It reaches the final answer much earlier (third and fourth subproblems). In the final answer, the solver simply

reiterates the steps that it earlier generated to answer the subproblems. This is a common behavior that we observed across multiple problems from multiple datasets. In Appendix D (see Figures 3 and 4), we provide similar case studies on LLaMA 13B and 33B with different prompting methods. With reduced solver capacity, the difference between Least-to-most and CoT generated reasoning steps further diminishes with both leading to incorrect answers; DaSLaM, on the other hand, still guides the solver through correct steps.

An interesting observation can be made by comparing how the solver behaves with CoT vs. with DaSLaM. With DaSLaM, we do not provide any new knowledge to the solver. Yet, the same model can rectify its errors made in CoT response. This may point to the intuition that current LLMs are actually underutilized, and one can unfold even more impressive performance with cleverly composed guidance.

We further provide case analyses with when DaSLaM fails to guide the model (GPT 3.5 in this case) to successful final answers, in Appendix E. While we do not find any obvious pattern of errors, one can see that the decomposer generates questions that are not readily answerable within that context. DaSLaM does not use any method to trace back the error or generate subproblems based on the answers to the previous subproblems. This might raise such issues where the subproblems generated are not actually helping the solver in the right order.

## 8   Conclusion

We challenged the design of ever-larger monolithic LLMs as homogeneous network structures, where diverse aspects of problem decomposition and solution are stored in a tangled and opaque manner. The formidable general-purpose problem-solving capabilities of LLMs are exceedingly resource-hungry, dependent on immense data engineering. Inspired by brain science, we took a first step toward heterogeneity — let two different LLMs evolve independently and adapt to their roles of decomposing and solving complex reasoning problems. Through extensive experiments on several benchmarks, we showed that such a heterogeneous network can match or exceed some of the largest contemporary LLMs, at a much smaller parameter count.

## Limitations

A potential limitation of DaSLaM, as with many system that uses an LLM-as-a-service API charging per token exchange, is the increased token usage because of the RL exploration. Asserting a token budget on the decomposer LM is left as an avenue for future exploration. Ideally, the decomposer LM should seamlessly invoke solvers of many forms, such as retrievers (Khattab et al., 2022b) or mathematical calculators (Schick et al., 2023; Wolfram, 2023). Future work may extend DaSLaM to such tools. DaSLaM is limited to purely text-based subproblem decomposition; it is not possible at present to incorporate reasoning through other modalities (e.g., visual inputs for geometric reasoning) into DaSLaM in its current form.

## Acknowledgments

The authors acknowledge the financial support of DYSL-AI.

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

# Small Language Models Fine-tuned to Coordinate Larger Language Models improve Complex Reasoning
## (Appendix)

## A  Supervised Fine-tuning Dataset

The training data for supervised fine-tuning stage was generated using `text-davinci-003`. Each data point in the dataset consisted of a tuple $\langle Q_{\text{gold}}, S_{\text{gold}}, Q'_{\text{gold}} \rangle$, where $Q_{\text{gold}}$ represents the reasoning question, $S_{\text{gold}}$ represents the gold reasoning steps, and $Q'_{\text{gold}}$ represent the sub-questions generated by `text-davinci-003` in a one-shot setting. An example of a single data point is given below.

---
**$Q_{\textbf{gold}}$**

John borrowed 3 soccer boots from Jake, and forgot them on the field, if peter came across a total of 15 boots on the field Jake's boots inclusive, and he took 4 boots at random, what is the probability that Jake's boots were not amongst the 4 taken?. (A) $\frac{12}{91}$, (B) $\frac{3}{15}$, (C) $\frac{12}{15}$ (D) $\frac{33}{91}$, (E) $\frac{3}{91}$

---
**$S_{\textbf{gold}}$**

Since Jake owns 3 of the boots, the subset from which the 4 boots should be chosen are the 12 boots not owned by Jake from the universe of 15.The first boot can be one of the 12 from the 15 with probability $\frac{12}{15}$. The second boot can be one of the 11 from the 14 remaining with probability $\frac{11}{14}$.The third boot can be one of the 10 from the 13 remaining with probability $\frac{10}{13}$. The fourth boot can be one of the 9 from the 12 remaining with probability $\frac{9}{12}$.The total probability will be $\frac{12}{15} \cdot \frac{11}{14} \cdot \frac{10}{13} \cdot \frac{9}{12}$ . On cancellation, this comes to $\frac{33}{91}$

---
**$Q'_{\textbf{gold}}$**

1. How many boots did Jake own?
2. How many boots were on the field?
3. How many boots did Peter take?
4. What is the probability of choosing one of the 12 boots not owned by Jake from the universe of 15?
5. What is the probability of choosing the second, third and fourth boots not owned by Jake?
6. What is the total probability?

---

## B  Reward Calculation

During policy gradient optimization of the decomposer LM $\phi$, rewards were provided in an incremental fashion so that the generated sub-questions align well with the initial question, the answer generated by the solver LM $\theta$ and the sub-question solving capabilities of $\theta$. For this, the reward was designed as a combination of five terms. The definitions of these terms are given in Section 4. Here we provide an example of the reward calculation.

*Entity coverage Reward ($R_1$).* Here, we find distinct nouns and numbers in the question and the sub-questions using the *nltk* library. The reward is calculated using $R_1 = \frac{|E_{\mathbf{Q}'}|}{|E_Q|}$, where $E_{\mathbf{Q}'}$ and $E_Q$ are the sets of distinct nouns and numbers in the sub-questions and questions, respectively. An example of the same is shown below.

---
$Q_{\text{gold}}$:
Each good worker can paint my new house alone in 12 hours. Each bad worker can paint my house alone in 36 hours. I need my house painted in 3 hours. If I can only find 3 good workers, how many bad workers must I also find in order to have my house painted on time?

$Q'_{\text{gold}}$:
1. How many good workers are needed to paint the house in 3 hours?
2. How many bad workers are needed to paint the house in 3 hours?
3. What is the total number of workers needed to paint the house in 3 hours?

$$|E_{Q'}| = 4 + 1 = 5 \tag{11}$$
$$|E_{Q'}| = 6 + 3 = 9 \tag{12}$$
$$R_1 = \frac{|E_{Q'}|}{|E_Q|} = \frac{5}{9} = 0.55 \tag{13}$$

---

*Consistency of answers to subproblems reward ($R_2$).* To ensure that the sub-question is such that the solver LLM $\phi$ does not produce random answers, we compare the entity in the solution to the entity asked in the subquestion along with the cosine similarity of the embeddings of sub-question and sub-answers. To find the embeddings, we use the pre-trained model in the *spacy* library. $R_2 = \sum_i (I(e_i = \hat{e}_i) + \text{cosine-similarity}(Q'_i, A_i))$, where $I(x) = 1$ if $x$ is true, $\hat{e}_i$ is entity whose value has been asked in the subproblem $Q'_i$, and $e_i$ is the entity answered. An example for the reward calculation is given below.

$Q'_1$: How many good workers are needed to paint the house in 3 hours?

$A_1$: We can calculate the amount of work done in 3 hours by a good worker by dividing 12 hours by 3 hours. We get 4 as the answer. We know that a bad worker can paint the house in 36 hours. We can calculate the amount of work done in 3 hours by a bad worker by dividing 36 hours by 3 hours. We get 12 as the answer. We need to find out how many good workers are needed to paint the house in 3 hours. We can calculate the number of good workers needed by dividing 12 by 4.

Answer: 3 good workers can paint the house in 3 hours.

cos-sim$(Q'_i, A_i) = 0.70457435$

$I(e_i = \hat{e}_i) = 1$

$R_2 = 1 + 0.70457435 = 1.70457435$

---

**Order of operations reward ($R_3$).** To evaluate the correctness of the mathematical solution produced by the model, we compare the order of operations. $R_3 = \frac{l}{m}$, where $l$ is the number of operations matched in order between $S$ and $S_{\text{gold}}$, and $m$ is the total number of operations in $S_{\text{gold}}$. Below is an example of $R_3$ calculation.

---

$S_{\text{model}}$:

Original Price = x

Discounted Price = 0.78x

Price after coupon = 0.78x - 20

Price paid = 0.78x - 20 + 1.90

0.78x - 20 + 1.90 = 0.5x

0.28x = 21.90

x = $\boxed{78.21}$

$S_{\text{gold}}$ : Let x be the original price of item

Discounted price = 0.78x

Payment made by the customer after using the $20 coupon = 0.78x - 20

0.78x - 20 = x/2 + 1.9

x = $\boxed{78.20}$

**operations in $S_{\text{model}}$**: [-, -, +, -, +]

**operations in $S_{\text{gold}}$**: [-, -, /, +]

$R_3 = \frac{3}{4} = 0.75$

---

**CoT Proximity reward ($R_4$).** As defined in Section 4, $R_4 = \sum_{j=0}^{m} I(c_{1j} > c_{2j})c_1 + I(c_{2j} > c_{1j})(-1 - c_{2j})$, where $I(\cdot)$ is the indicator function. Here we present an example to illustrate the same.

---

$S_{gold}$

1. : We first consider the possible neighbors of the chief's mother, one must be the chief, the other is one of the remaining 5 natives. That native can, in turn, have one of 4 neighbors in addition to the chief's mother

2. Continuing around, there are 5! = 120 possible arrangements

3. The chief can be on his mother's left or right, for a total of $2 \cdot 120 = 240$ arrangements

4. Alternatively, we know that the seating arrangement is the same if it is rotated, once the chief's spot is chosen, the rotations are eliminated

5. There are 2 ways to seat his wife and his mother, and then there are 5 distinct positions for the remaining 5 natives, which makes for $2 \cdot 5! = \boxed{240}$ possible arrangements

$S_0$:

1. We can see that the chief must be seated between his mother and his wife.

2. This means that there are 3 possible positions for the chief, and 5 possible positions for the other 6 people.

3. Therefore, there are $3! \cdot 5! = \boxed{1800}$ different seating arrangements.

$S$:

1. The chief must sit between his mother and his wife.

2. There are 8 people in the tribe.

3. So, the chief's mother and wife can be arranged in 2! ways.

4. The remaining 5 people can be arranged in 5! ways.

5. Therefore, the total number of possible seating arrangements after removing rotation = 5! = $\boxed{240}$

Cosine similarity calculation per step:

1. $c_{1j} = 0.47673503, c_{2j} = 0.44773823$
2. $c_{1j} = 0.45063934, c_{2j} = 0.47917843$
3. $c_{1j} = 0.5173945, c_{2j} = 0.20383504$
4. $c_{1j} = 0.46866685, c_{2j} = 0$
5. $c_{1j} = 0.47825924, c_{2j} = 0$

Hence, $R_4 = 0.47673503 + (-1 - 0.47917843) + 0.5173945 + 0.46866685 + 0.47825924 = 0.46187719$

---

**Correctness of final answer($R_5$),** $R_5 = I(\hat{A} = A_{\text{gold}})$. This reward checks if the final answer matches the gold answer. A negative example of the same is given below.

$Q_{gold}$:
Three friends Alan, Roger and Peter attempt to answer a question on an exam. Alan randomly guesses the answer, giving him a $\frac{1}{5}$ probability of guessing correctly. Roger cheats by looking at the paper of the student in front of him, giving him a $\frac{2}{3}$ probability of answering correctly. And Peter dutifully performs the calculations, then marks the answer, giving him a $\frac{5}{6}$ probability of a correct answer. What is the probability that the question is answered correctly, but not via cheating?

$S_{\text{gold}}$ :
Prob(Alan) = $\frac{1}{5}$
Prob(Roger) without cheating = $\frac{2}{3} - 1 = \frac{1}{3}$
Prob(Peter) = $\frac{5}{6}$

Total Probability = $\frac{1}{5} \cdot \frac{1}{3} \cdot \frac{5}{6} = \boxed{\frac{1}{18}}$

$S_{\text{model}}$ :
Alan has a $\frac{1}{5}$ chance of getting the answer correct.
Roger has a $\frac{2}{3}$ chance of getting the answer correct.
Peter has a $\frac{5}{6}$ chance of getting the answer correct.
The probability that the question is answered correctly is $\frac{1}{5} + \frac{2}{3} + \frac{5}{6} = \frac{13}{12}$.
The probability that the question is answered correctly, but not via cheating is $1 - (\frac{1}{5} + \frac{2}{3} + \frac{5}{6}) = 1 - \frac{13}{12} = \boxed{\frac{-1}{12}}$

$$A_{\text{gold}}: \frac{1}{18}$$
$$A_{\text{model}}: \frac{-1}{12}$$
$$R_5 = 0$$

## C  Hyperparameter Selection

We performed hyperparameter tuning on a subset of 250 examples from the training dataset. Based on the improvement in accuracy and compute resources, we converged at the following values.

For the supervised fine-tuning stage, we used LoRA $r = 16$, LoRA $\alpha$ = 32, LoRA droput = 0.05. For the RLMF stage, we finetuned the last 3 layers of LoRA adapters, using a batch size of 8, gradient accumulation steps=8, init kl coef=0.01, target=4. For inference, we used the following generation parameters: temperature = 0.95, top p=0.18, pad token id = 0, do sample = False, number of beams = 1, maxi length = 2048.

All the models were implemented using Huggingface with PyTorch, and loaded in int_8 precision. For solver-LLaMA 13B model, we instruction finetuned the LLaMA13B model using the Alpaca dataset by stanford (Taori et al., 2023). For solver-LLaMA 33 billion model, we used an instruction-finetuned version of LLAMA-33 billion from Huggingface. To implement the PPO algorithm, we used the TRL library. We used a single NVIDIA A100 (80GB) GPU for all training and inference purposes.

## D  Case Study on LLAMA Models

Similar to Figure 2, we performed example-wise case study on the answering process of the LLaMA solvers (13 billion and 33 billion models) with different prompting methods (see Figures 3 and 4). The example problem was selected from the MATH dataset. As we can see, with relatively small scale solvers like LLaMA, Least-to-most fails to add any improvement over basic CoT prompting. DaSLaM is able to guide the model to the correct answer in both cases.

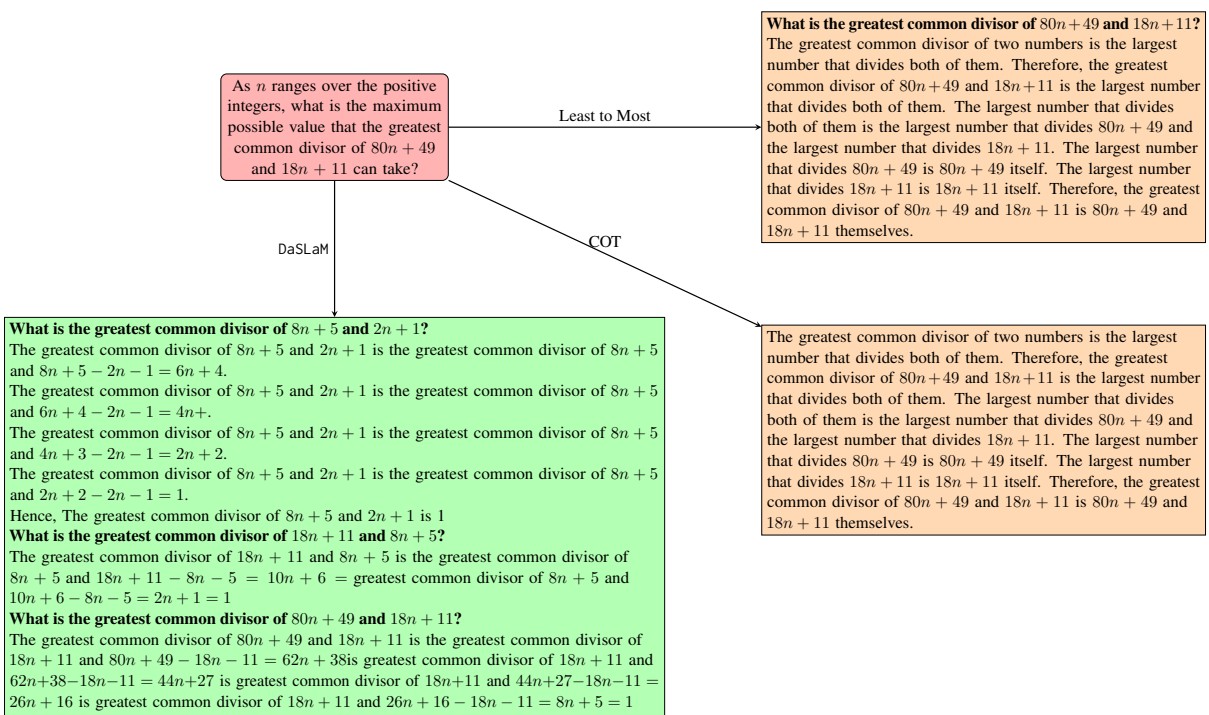

Figure 3: A case study on LLAMA-13B.

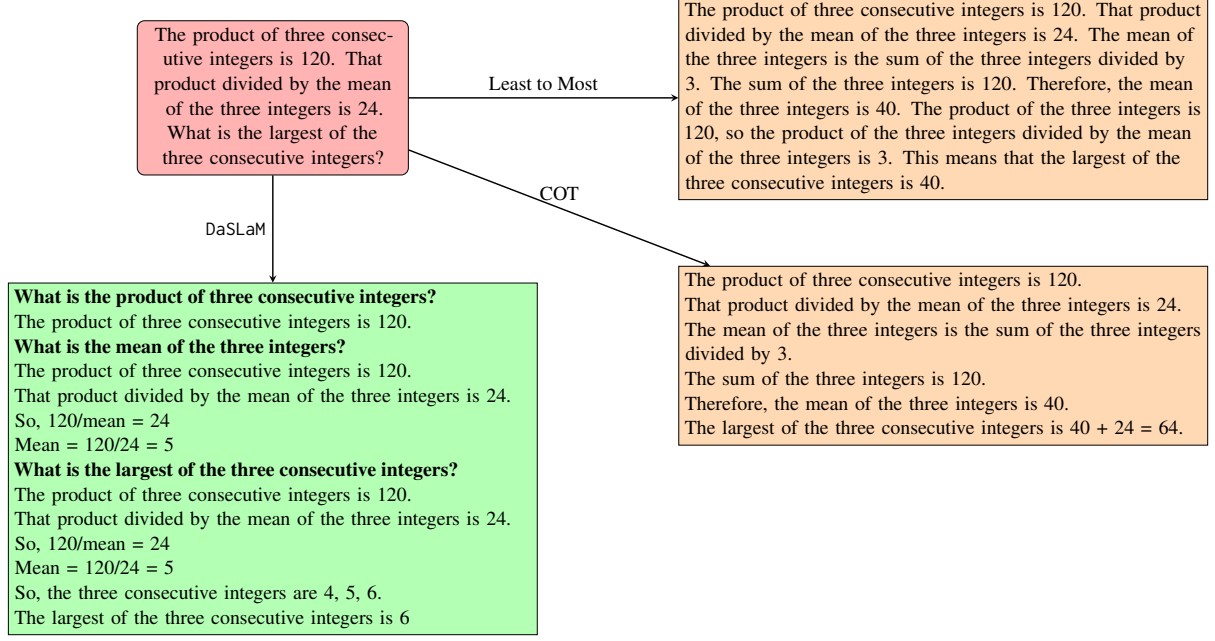

Figure 4: A case study on LLaMA-33B as solver model.

## E Failure cases with `DaSLaM`

### E.1 Case I

**Question** A train running at a speed of 100 miles/hour, takes 10 hours to reach its destination. After covering quarter of the distance, it starts raining and the train has to be slowed to speed of 75 miles/hour. What is the total journey duration? A)10 B)11.5 C)12.5 D)13.5 E)15

**Initial Answer** The train covers the first quarter of the distance at a speed of 100 miles/hour.
So, the time taken to cover the first quarter of the distance = (1/4) x 10 hours = 2.5 hours
The train covers the remaining three quarters of the distance at a speed of 75 miles/hour.
So, the time taken to cover the remaining three quarters of the distance = (3/4) x 10 hours = 7.5 hours
Therefore, the total journey duration = 2.5 + 7.5 = 10 hours

**Sub problem 1** What is the total distance to be covered?

**Sub Answer 1** Let the total distance to be covered be d.
Time taken to cover the first quarter of the distance = (1/4)d/100 = 10/400 hours
Time taken to cover the remaining distance = (3/4)d/75 = 300/400 hours
Total time taken = 10/400 + 300/400 = 10/4 hours = 2.5 hours
Since the train takes 10 hours to reach its destination,
Total distance to be covered = 10 x 100 = 1000 miles

**Sub problem 2**
What is the speed of the train?

**Sub Answer 2**
After covering quarter of the distance, the train has to be slowed to speed of 75 miles/hour.
Let the remaining distance be x.
Therefore, x = d - (d/4) = 1000 - (1000/4) = 750 miles.
The time taken to cover the remaining distance x = 750 miles at a speed of 75 miles/hour = (750/75) hours = 10 hours.
Therefore, the speed of the train = x/time taken = 750/10 = 75 miles/hour.

**Sub problem 3** What is the time taken to cover the distance?
**Sub Answer 3**
Time taken to cover the first quarter of the distance = (d/4)/100 hours = d/400 hours
Time taken to cover the remaining distance = (3d/4)/75 hours = d/300 hours
Total time taken = d/400 + d/300 = (5d + 4d)/1200 hours = 9d/1200 hours.
Putting d=1000,
Final answer: The time taken to cover the distance is 9 x 1000/1200 hours.
**Sub problem 4** What is the total duration of the journey?
**Sub Answer 4** Time taken to cover the first quarter of the distance = (d/4)/100 hours = d/400 hours
Time taken to cover the remaining three quarters of the distance = (3d/4)/75 hours = 4d/300 hours
Total time taken = d/400 + 4d/300 = (5d + 4d)/(400 + 300) = 9d/700 hours
Total duration of the journey = 9d/700 hours = 9 hours and (d/700) hours = 9 hours and (100/700) hours = 9 hours and 14.29 minutes

**Final Answer** Step 1: The train covers the first quarter of the journey at a speed of 100 miles/hour. Step 2: The time taken to cover the first quarter of the journey is given by: Time = Distance/Speed Time = (1/4) x 100/100 Time = 1/4 hour Step 3: The train covers the remaining three quarters of the journey at a speed of 75 miles/hour. Step 4: The time taken to cover the remaining three quarters of the journey is given by: Time = Distance/Speed Time = (3/4) x 75/100 Time = 3/4 hour Step 5: The total duration of the journey is the sum of the time taken to cover the first quarter and the remaining three quarters of the journey. Total Time = 1/4 + 3/4 Total Time = 4/4 Total Time = 1 hour Therefore, the total duration of the journey is 1 hour.

## E.2 Case II

**Question** Consider there is an staircase elevator and you are coming down. If you walk 20 steps and stop, then you reach bottom in 10 minutes. If you walk 10 steps and stop, you reach to the ground in 20 minutes. What is the speed of the elevator? A)1 step/minute B)2 step/minute C)3 step/minute D)4 step/minute E)None of the above

**Initial Answer** .

**Sub Question 1** What is the total number of steps in the elevator?

.

**Sub Question 2** What is the speed of the elevator?

.

**Sub Question 3**

What is the time taken to cover 20 steps?

**Sub Answer 3**

Time taken to cover 20 steps is 10 minutes

**Sub Question 4**

What is the time taken to cover 10 steps?

**Sub Answer 4**

Time taken to cover 10 steps is 20 minutes

**Final Answer** Let the speed of the elevator be x.
Time taken to reach the bottom when walking 20 steps and stopping = 10 minutes
Time taken to reach the bottom when walking 10 steps and stopping = 20 minutes
Therefore,
20x = 10
10x = 20
x = 2
Therefore, the speed of the elevator is 2 steps/minute.

## E.3 Case III

**Question** Let $-\frac{\pi}{6} < \theta < -\frac{\pi}{12}$. Suppose $\alpha_1$ and $\beta_1$ are the roots of the equation $x^2 - 2x \sec \theta + 1 = 0$ and $\alpha_2$ and $\beta_2$ are the roots of the equation $x^2 + 2x \tan \theta - 1 = 0$. If $\alpha_1 > \beta_1$ and $\alpha_2 > \beta_2$, then $\alpha_1 + \beta_2$ equals

  (A) $2(\sec \theta - \tan \theta)$
  (B) $2 \sec \theta$
  (C) $-2 \tan \theta$
  (D) $0$

**Initial Answer**

We can solve the equations for $alpha_1$ and $alpha_2$ to get

  $alpha_1 = \sec \theta + \tan \theta$
  $alpha_2 = -\sec \theta + \tan \theta$
  Since $\alpha_1 \beta_1$ and $\alpha_2 \beta_2$, we have
  $\sec \theta + \tan \theta > -\sec \theta$
  $\sec \theta + \tan \theta > -\tan \theta$
  Adding the two equations, we get
  $2 \sec \theta > 0$
  Therefore, $\alpha_1 \beta_2 = 2 \sec \theta$.
  The correct answer is (B) $2 \sec \theta$.

**What is the equation of the line** $x^2 - 2x \sec \theta + 1 = 0$**?**

Given, $-\frac{\pi}{6} < \theta < -\frac{\pi}{12}$ We need to find the equation of the line $x^2 - 2x \sec \theta + 1 = 0$ We know that the equation of a line is given by $y = mx + c$ Now, let us substitute the given equation in the equation of a line. $x^2 - 2x \sec \theta + 1 = 0$
$y = mx + c$
$x^2 - 2x \sec \theta + 1 = mx + c$
$mx + c - x^2 + 2x \sec \theta - 1 = 0$
$mx - x^2 + 2x \sec \theta = 1$
Comparing the coefficients of $x$ on both sides,
$m = 2 \sec \theta$
Comparing the constant terms on both sides,
$c = 1$
Therefore, the equation of the line $x^2 - 2x \sec \theta + 1 = 0$ is given by
$y = 2 \sec \theta x + 1$
Hence, the answer is $y = 2 \sec \theta x + 1$.

**What is the equation of the line** $x^2 + 2x \tan \theta - 1 = 0$**?**

Given, $-\frac{\pi}{6} < \theta < -\frac{\pi}{12}$
We know that,
$\sec \theta = \frac{1}{\cos \theta}$
$\tan \theta = \frac{\sin \theta}{\cos \theta}$
Therefore,
$x^2 - 2x \sec \theta + 1 = x^2 - 2x \frac{1}{\cos \theta} + 1 = 0$
$x^2 + 2x \tan \theta - 1 = x^2 + 2x \frac{\sin \theta}{\cos \theta} - 1 = 0$

Hence, the equation of the line $x^2 + 2x \tan \theta - 1 = 0$ is $x^2 + 2x \frac{\sin \theta}{\cos \theta} - 1 = 0$.

Final Answer: $x^2 + 2x \frac{\sin \theta}{\cos \theta} - 1 = 0$

**What is the equation of the line $x^2 - 2x \sec \theta + 1 = 0$ minus the equation of the line $x^2 + 2x \tan \theta - 1 = 0$?**

The equation of the line $x^2 - 2x \sec \theta + 1 = 0$ is given by $x^2 - 2x \sec \theta + 1 = 0$

The equation of the line $x^2 + 2x \tan \theta - 1 = 0$ is given by $x^2 + 2x \tan \theta - 1 = 0$

Subtracting the two equations, we get

$x^2 - 2x \sec \theta + 1 - x^2 - 2x \tan \theta + 1 = 0$

Simplifying the equation, we get

$-2x(\sec \theta + \tan \theta) = 0$

Therefore, the equation of the line $x^2 - 2x \sec \theta + 1 = 0$ minus the equation of the line $x^2 + 2x \tan \theta - 1 = 0$ is $-2x(\sec \theta + \tan \theta) = 0$

Hence, the answer is $-2x(\sec \theta + \tan \theta) = 0$.

**What is the equation of the line $x^2 - 2x \sec \theta + 1 = 0$ plus the equation of the line $x^2$**

Given, $-\frac{\pi}{6} < \theta < -\frac{\pi}{12}$

$\alpha_1$ and $\beta_1$ are the roots of the equation $x^2 - 2x \sec \theta + 1 = 0$

$\alpha_2$ and $\beta_2$ are the roots of the equation $x^2 + 2x \tan \theta - 1 = 0$

$\alpha_1 > \beta_1$ and $\alpha_2 > \beta_2$

We need to find the value of $\alpha_1 + \beta_2$

Let us consider the equation $x^2 - 2x \sec \theta + 1 = 0$

$\alpha_1$ and $\beta_1$ are the roots of the equation.

Therefore,

$\alpha_1 + \beta_1 = -2 \sec \theta$

Let us consider the equation $x^2 + 2x \tan \theta - 1 = 0$

$\alpha_2$ and $\beta_2$ are the roots of the equation.

Therefore,

$\alpha_2 + \beta_2 = 2 \tan \theta$

Now,

$\alpha_1 + \beta_2 = \alpha_1 + \beta_2 + \alpha_2 + \beta_2 - \alpha_2 - \beta_2$

$= \alpha_1 + \beta_1 - \alpha_2 - \beta_2$

$= -2 \sec \theta + 2 \tan \theta$

$= 2(\sec \theta - \tan \theta)$

Hence, the answer is (A) $2(\sec \theta - \tan \theta)$.

$\alpha_1 > \beta_1$ and $\alpha_2 > \beta_2$

We need to find the value of $\alpha_1 + \beta_2$

Let us consider the equation $x^2 - 2x \sec \theta + 1 = 0$

$\alpha_1$ and $\beta_1$ are the roots of the equation.

Therefore,

$\alpha_1 + \beta_1 = -2 \sec \theta$

Let us consider the equation $x^2 + 2x \tan \theta - 1 = 0$

$\alpha_2$ and $\beta_2$ are the roots of the equation.

Therefore,

$\alpha_2 + \beta_2 = 2 \tan \theta$

Now,

$\alpha_1 + \beta_2 = \alpha_1 + \beta_2 + \alpha_2 + \beta_2 - \alpha_2 - \beta_2$

$= \alpha_1 + \beta_1 - \alpha_2 - \beta_2$

$= -2 \sec \theta + 2 \tan \theta$

$= 2(\sec \theta - \tan \theta)$

Hence, the answer is (A) $2(\sec \theta - \tan \theta)$.

**Final Answer** Given,

$-\frac{\pi}{6} < \theta < -\frac{\pi}{12}$

$\alpha_1$ and $\beta_1$ are the roots of the equation $x^2 - 2x \sec \theta + 1 = 0$

$\alpha_2$ and $\beta_2$ are the roots of the equation $x^2 + 2x \tan \theta - 1 = 0$