# OpenReview forum: "Small Language Models Fine-tuned to Coordinate Larger Language Models improve Complex Reasoning"
_EMNLP/2023/Conference — EMNLP 2023 Main_

### Official Review · Reviewer_xvjb · 2023-08-02

**Soundness:** 4

**Excitement:**

3: Ambivalent: It has merits (e.g., it reports state-of-the-art results, the idea is nice), but there are key weaknesses (e.g., it describes incremental work), and it can significantly benefit from another round of revision. However, I won't object to accepting it if my co-reviewers champion it.

**Missing References:**

[1] Successive Prompting for Decomposing Complex Questions, Dua et al. 2022
[2] Parsel: Algorithmic Reasoning with Language Models by Composing Decompositions, Zelikman et al. 2022

**Paper Topic And Main Contributions:**

This paper proposes to split up LM problem solving with two separately finetuned LMs, one for decomposition and one for solving subproblems.

**Questions For The Authors:**

See weaknesses

**Reasons To Accept:**

The idea is fairly natural and simple to motivate. The paper is well-presented and compelling.

**Reasons To Reject:**

The comparisons feel quite unfair. The approach fine-tunes their model, while the alternatives are all exclusively prompting-based. To me, the most important ablation to include would have been one where the model is asked to decompose and solve the problems without training. In my eyes, the main question that this paper has to answer is not whether decomposition helps in answering reasoning questions - much prior work (e.g. [1-3]) has already shown this. As indicated by [2], sometimes knowing what question to ask next requires an answer to a previous question - however, as far as I understand, the approach in this paper has no way of handling that (though that is not a reason to reject it in itself - not every paper has to solve every problem - but it weakens the message).

In other words, the main contribution of the paper is not that decomposing and solving helps, but that this can be improved with fine-tuning as they propose. However, as far as I understand, this paper does not perform this analysis which is central to the work.

1. [1] Successive Prompting for Decomposing Complex Questions, Dua et al. 2022
2. [2] Demonstrate-Search-Predict: Composing retrieval and language models for knowledge-intensive NLP, Khattab et al. 2022
3. [3] Parsel: Algorithmic Reasoning with Language Models by Composing Decompositions, Zelikman et al. 2022

**Reproducibility:**

3: Could reproduce the results with some difficulty. The settings of parameters are underspecified or subjectively determined; the training/evaluation data are not widely available.

**Reviewer Confidence:**

4: Quite sure. I tried to check the important points carefully. It's unlikely, though conceivable, that I missed something that should affect my ratings.

---

> ### Author Rebuttal · Authors · 2023-08-29
>
> Thank you for your constructive criticism of our paper. Following are the responses to the concerns you raised under **Reasons to reject**:
>
> > The comparisons feel quite unfair. The approach fine-tunes their model, while the alternatives are all exclusively prompting-based. To me, the most important ablation to include would have been one where the model is asked to decompose and solve the problems without training.
>
> We emphasize that **the solver model is not finetuned in any of the experiments**. Only the decomposer is finetuned. Since this is a LLaMA 13B model, our preliminary experiments suggest that the question generation capability of the decomposer **before** finetuning is very poor, as shown below:
>
> | Method | PnC | NT | ALG | iALG | Geom. | Cal | Palg | AQuA |
> | -------- | -------- | -------- | -------- | -------- | -------- | -------- | -------- | -------- |
> | DaSLaM (text-davinci-003)     | 21.4    | 26.1    | 33.4     | 24.8     | 18.2    | 44.0    | 21.4     | 54.5 |
> | DaSLaM (text-davinci-003) prompted decomposer     | 12.4 | 9.6 | 23.2 | 7.6 | 8.33 | 17.8 | 8.1 | 39.0 |
> | DaSLaM (LLaMA 13B)     | 4.0    | 5.6    | 4.70     | 3.4     | 4.3    | 4.1    | 8.33     | 22.0 |
> | DaSLaM (LLaMA 13B) prompted decomposer     | 3.8 | 3.83 | 4.6 | 3.2 | 3.66 | 3.05 | 5.0 | 19.6 |
> | DaSLaM (LLaMA 33B)     | 4.0 |  7.36 | 9.09 | 6.02 | 5.3 | 6.03 | 8.44 | 26.8     |
> | DaSLaM (LLaMA 33B) prompted decomposer     | 1.6 | 3.2 | 5.6 | 4.7 | 3.2 | 2.4 | 6.66 | 21.6 |
>
> As we can see, there is >50% reduction in accuracy in most tasks when we replace the finetuned decomposer with a prompted one.
>
> We claim that the model-level separation of the decomposition and solution steps in DaSLaM is superior to single-model decomposition-solution approaches like [1], irrespective of the power of the solver. We used Least-to-most prompting as a baseline to validate this claim.
>
> > sometimes knowing what question to ask next requires an answer to a previous question - however, as far as I understand, the approach in this paper has no way of handling that
>
> While this is not the central question of our paper, our proposed method partially addresses this. While [1] seeks to generate a subproblem based on the original problem and the answer to the subproblem, they do not take into account how the solver acts against the original question. We showed that this feedback plays a crucial role; the decomposer in our ablation variant DaSLaM-NF generates the subproblem independent of the initial response of the solver. This ablation variant performs pretty poorly, irrespective of the decomposer finetuning.
>
> > the main contribution of the paper is not that decomposing and solving helps, but that this can be improved with fine-tuning as they propose
>
> We would like to point out that our contribution is not just limited to finetuning. A major bottleneck with finetuning is that most existing LLMs are either prohibitively large to finetune using a constrained resource or the API providers like OpenAI do not provide such a facility. On the other hand, the superiority of such large models as solvers compared to (even finetuned) smaller models is generally accepted. Our method provides a novel way to incorporate finetuning-based knowledge transfer into such large models while treating them in a black-box manner.
>
> > Missing references
>
> We commit to include discussions on the mentioned papers in the revised version.
>
> > Reproducibility: The settings of parameters are underspecified or subjectively determined; the training/evaluation data are not widely available.
>
> We have provided the full codebase and datasets used in the Supplementary along with the hyperparameter details in Appendix C.
>
> We commit to include the additional experimental results presented in the response in the revised version of our paper.
>
> [1] Dua, Dheeru, et al. "Successive prompting for decomposing complex questions." arXiv preprint arXiv:2212.04092 (2022).

---

### Official Review · Reviewer_fnE5 · 2023-08-04

**Soundness:** 4

**Excitement:**

4: Strong: This paper deepens the understanding of some phenomenon or lowers the barriers to an existing research direction.

**Missing References:**

See "Questions for The Authors".

**Paper Topic And Main Contributions:**

This paper proposes a method to train a small (e.g. 13B) *decomposition* model that guides step-by-step problem-solving by a black-box *solver* model (e.g., GPT-3.5) via reinforcement learning (RL). Unlike recent methods which use prompt-based methods to explicitly guide problem solving, this work uses RL to deeply tune a decomposition model to guide and thus improve the performance of the solver model. The results are very strong, demonstrating notable improvement over baseline prompting methods, and show clear generalization to other solver models (e.g., LlaMA) and harder benchmarks (e.g., JEE) unseen during training.

**Questions For The Authors:**

- E. Is this the first paper to employ RL to train an auxiliary model to guide complex reasoning, or other forms of problem solving, by very large LLMs? This is not explicitly stated in the paper, and if there are other papers, discussion of these would help readers better understand the paper's contribution.

**Reasons To Accept:**

- The authors propose a novel approach to apply RL to deeply tune mid-sized LLMs to guide complex reasoning (of much larger LLMs). The proposed approach differentiates itself from previous works from a technical standpoint, as these are based on prompting.
- The paper tackles the important issue of hallucinations and errors in complex reasoning, highlighted by the abundance of recent work. Relevant literature appears to be thoroughly discussed throughout the paper, but I am not certain.
- The method shows remarkable performance improvements over prompt-based baselines and are shown to generalize well to novel solver models and benchmarks.
- The method is applicable to black-box solver LLMs, which is the case for recent state-of-the-art models such as GPT-4 and PaLM-2.

**Reasons To Reject:**

- A. The motivation proposed by the authors that "... decomposition and solution generation .... need distinct capabilities" (Line 13) is not sufficiently supported. The citation from the neuroscience literature in Line 67 ("cerebral functions are specialized and spatially localized") does not support the need for a separate language models for decomposition and solution generation, for two reasons: (1) the connection assumes that language models and human brains work in the same way, which is risky conjecture and (2) multiple cognitive abilities of language models can also be localized in terms of model parameters, without the need for separate models–in fact, pre-trained language models already demonstrate a wide variety of abilities. On the contrary, concurrent work [1] shows that step-by-step supervision of a monolithic LLM can greatly improve complex reasoning performance, without the need for specialized models.
  - A1. In my opinion, the paper does not need to include this speculative motivation. The well-known issue of {hallucinations/errors from chain-of-thought reasoning, and the limitations of existing prompting methods to mitigate this issue} provide sound motivation to apply more explicit forms of supervision via RL to guide the model. The motivation to train a separate external model could come from the fact that foundational LLMs are typically not available for fine-tuning and it is computationally prohibitive to do so. In fact, the versatility of the proposed method, which only requires fine-tuning of a *small* language model and can be applicable to *black-box* models for the underlying solver model, is a major strength of the proposed method. This could be highlighted further in the introduction.
- B. The case study (Section 7) only pertains to one sample, suggesting that it may have been cherry-picked. While this is effective at highlighting the strength of the proposed method, overall qualitative analysis based on a number of random samples would help readers to understand the overall strengths and weaknesses. This could also be supported with more samples provided in the paper, highlighting both success and failure cases of the method. Comprehensive analysis of both strengths and weaknesses would greatly improve the paper.
- C. If results are available, it would be helpful for the readers to provide a full performance table for GPT-4 baselines (near Line 515), even if the results may not be favorable for the proposed method.
- D. The writing is generally easy to follow, but off-putting or erroneous at times. It is evident that the writing could benefit from more meticulous attention. I would encourage the authors to revise the text for a more polished and refined presentation, given the significant contribution of their work. A non-exhaustive list of potential issues are noted in "Typos Grammar Style And Presentation Improvements".
  - D1. The organization and writing for page 6 are notably very good.

I'd like to note that the idea and contribution of the paper are quite strong, but my initial scores will have to reflect the issues above. I am open to adjusting my score if these issues are addressed.

[1] Lightman 2023, Let's Verify Step by Step

**Reproducibility:**

4: Could mostly reproduce the results, but there may be some variation because of sample variance or minor variations in their interpretation of the protocol or method.

**Reviewer Confidence:**

5: Positive that my evaluation is correct. I read the paper very carefully and I am very familiar with related work.

**Typos Grammar Style And Presentation Improvements:**

- There is a small error in terminology regarding OpenAI models. `text-davinci-003` is also a variant of GPT 3.5 rather than GPT 3 (Line 108), according to OpenAI (https://platform.openai.com/docs/models/gpt-3-5).
- In the caption of Figure 1, it is stated at "In step 1, the solver ... fails to answer correctly", but this is not always the case. The reader may ask, "what happens when it answers correctly?". The explanation could be improved by clarifying that the solver LM provides an initial answer, which is *often* found to have errors, for which the decomposition model takes into account when generating subproblems.
- In Line 171 ("the decomposition actuates with complete knowledge of the solver's actions"), I was confused by the phrase *complete knowledge*. A more appropriate explanation may be, e.g., that the decomposition is informed by the actions of the solver.
- The text explanations and equations in Section 3 could both be better polished and organized to enhance readability. I had/have many confusions including: (1) whether the decomposition model (DM) uses the entire chain-of-thought reasoning from the solver model (SM) or just the first step, as in Figure 1, (2) that the DM generates all subproblems in a single step, and the SM answers each of these independently, (3) whether the solver uses the solutions from previous subproblems to answer the current subproblem, (4) what is the difference between $\hat{A}_i$ and $S_i$ in Eq 4, i.e., "is the solver prompted to infer a final answer based on the local solution of each subproblem, at *each* step?", etc.
  - The mathematical expression `argmax [ p(...) argmax(...) ]` used throughout Section 3 is not standard syntax, therefore can be confusing.

---

> ### Author Rebuttal · Authors · 2023-08-29
>
> Thank you for your constructive criticism of our paper. Following are the responses to the concerns you raised under **Reasons to reject** and **Questions For The Authors**:
>
> > ("cerebral functions are specialized and spatially localized") does not support the need for separate LMs for decomposition and solution [...] (1) the connection assumes that LMs and human brains work in the same way [...] (2) multiple cognitive abilities of LMs can also be localized in terms of model parameters, without the need for separate models [...] step-by-step supervision of a monolithic LLM can greatly improve complex reasoning [...] without the need for specialized models.
>
> As you correctly point out, the neuroscience reference is not essential to develop our ideas, and will be removed.  However, from (1) above, you might have misunderstood our intent. We need not assume that LMs and human brains work in the same way (we know for sure they do not) to want to try something that works on humans on a LM.  CoT and RLHF work with both human brains and LLMs, after all! As for (2) above, we regard localization and modular editing research to be in nascent stages with much left to explain and invent.
>
> > paper does not need to include this speculative motivation. [...] The motivation to train a separate external model could come from the fact that foundational LLMs are typically not available for fine-tuning and it is computationally prohibitive to do so. In fact, the **versatility of the proposed method**, which only requires fine-tuning of a small language model and can be applicable to black-box models for the underlying solver model, is a **major strength of the proposed method**. This could be highlighted further in the introduction.
>
> Thanks! This sounds *much* better than we ourselves pitched the work! We will update our manuscript accordingly.
>
> >  The case study (Section 7) only pertains to one sample, suggesting that it may have been cherry-picked. While this is effective at highlighting the strength of the proposed method, overall qualitative analysis based on a number of random samples would help readers to understand the overall strengths and weaknesses. This could also be supported with more samples provided in the paper, highlighting both success and failure cases of the method.
>
> We greatly appreciate this suggestion. We included only one example due to space constraint in the main paper. We commit to include more examples from different datasets in the revised version, highlighting *both* the strengths and the weaknesses of our proposed method.
>
> > If results are available, it would be helpful for the readers to provide a full performance table for GPT-4 baselines
>
> Results of GPT-4 on the JEEBench dataset are available in [1]. We will include them under Table 3 in the revised version. For the MATH dataset, we could only find the aggregate results and not the topic-specific ones.
>
> > The writing is generally easy to follow, but off-putting or erroneous at times. It is evident that the writing could benefit from more meticulous attention. I would encourage the authors to revise the text for a more polished and refined presentation, given the significant contribution of their work.
>
> We greatly appreciate the suggestion. We commit to revising the current version and correcting such mistakes wherever present.
>
> > Is this the first paper to employ RL to train an auxiliary model to guide complex reasoning, or other forms of problem solving, by very large LLMs? This is not explicitly stated in the paper, and if there are other papers, discussion of these would help readers better understand the paper's contribution.
>
> To the best of our knowledge, ours is **the first such proposal**. Training verifier-like auxiliary models is not new, but they do not *actively* guide the solver along the complex reasoning requirements like the DaSLaM decomposer.
>
> [1] Arora, Daman, and Himanshu Gaurav Singh. "Have LLMs Advanced Enough? A Challenging Problem Solving Benchmark For Large Language Models." arXiv preprint arXiv:2305.15074 (2023)

---

### Official Review · Reviewer_KUis · 2023-08-05

**Soundness:** 4

**Excitement:**

3: Ambivalent: It has merits (e.g., it reports state-of-the-art results, the idea is nice), but there are key weaknesses (e.g., it describes incremental work), and it can significantly benefit from another round of revision. However, I won't object to accepting it if my co-reviewers champion it.

**Paper Topic And Main Contributions:**

Language Models Separately Tuned for Decomposition and Solution Improve Complex Reasoning

In this paper the authors suggest training a separate LLM that learns to decompose a problem in order to improve reasoning in LLMs. They do this by first finetuning the LM to produce the right subproblems. Then they use reinforcement learning to sensitize the decomposer to errors made by the solver. They show that this approach works better than other baselines such as chain of thought prompting in many cases.

**Reasons To Accept:**

+ addresses an important problem
+ observes that simpler models could produce better results by fine tuning---not a major insight, but still important to note
+ It looks like the results on JEEBench are good, which suggests some capacity for generalization for the decomposer model.

**Reasons To Reject:**

- some aspects of the description was not clear, such as the cosine similarity which seems to use a different model
- only prompting methods used as baselines. The approach uses detailed finetuning with extra architectural elements and tuning to solvers but no baselines contain any elements similar to these. It is not clear from the experiments if finetuning a baseline could also yield the benefits shown.
- The approach requires knowing solver errors and annotating with decompositions to tune the decomposer. This seems like a limitation, since that would require significant after-the-fact human effort in most domains especially with more complex answers than multiple choice or simple numeric.

**Reproducibility:**

3: Could reproduce the results with some difficulty. The settings of parameters are underspecified or subjectively determined; the training/evaluation data are not widely available.

**Reviewer Confidence:**

3: Pretty sure, but there's a chance I missed something. Although I have a good feel for this area in general, I did not carefully check the paper's details, e.g., the math, experimental design, or novelty.

---

> ### Author Rebuttal · Authors · 2023-08-29
>
> Thank you for your constructive criticism of our paper. Following are the responses to the concerns you raised under **Reasons to reject**:
>
> > only prompting methods used as baselines. The approach uses detailed finetuning with extra architectural elements and tuning to solvers but no baselines contain any elements similar to these. It is not clear from the experiments if finetuning a baseline could also yield the benefits shown.
>
> Our proposed approach DaSLaM **does not finetune** the solver model. It only finetunes the decomposer model (LLAMA 13B) to generate subquestions based on the output of the solver. Yet, as per the comment, we have recently experimented with full finetuning of an LLAMA 13B model on the training dataset we used for tuning the decomposer; the results are shown below:
>
>
> | Method | PnC | NT | ALG | iALG | Geom. | Cal | Palg | AQuA |
> | -------- | -------- | -------- | -------- | -------- | -------- | -------- | -------- | -------- |
> | DaSLaM (LLaMA 13B)     | 4.0    | 5.6    | 4.70     | 3.4     | 4.3    | 4.1    | 8.33     | 22.0     |
> | Finetuned (LLaMA 13B)    | 2.8   | 3.6   | 3.57    | 3.2  | 4.1     | 3.05   | 6.04  | 19.4  |
>
> Clearly, despite using a non-tuned solver, DaSLaM delivers better performance compared to full finetuning. Even though the decomposer is finetuned, it is solver-agnostic. So, it can be used with different solvers (as we have shown in the paper with three different solvers). Furthermore, it is prohibitive to finetune most existing LLMs. Our method provides a strong solution to this problem.
>
> > The approach requires knowing solver errors and annotating with decompositions to tune the decomposer. This seems like a limitation, since that would require significant after-the-fact human effort in most domains especially with more complex answers than multiple choice or simple numeric.
>
> There is a misunderstanding. DaSLaM **does not require** a step-by-step error annotation of the solver output. While training the decomposer, we only need whether the final output from the solver is correct or not as supervision signal. The whole point of using RL to train the decomposer is to bypass such costly human intervention using simple, automated rewards computed from the existing chain-of-thought annotated datasets.
>
> > some aspects of the description was not clear, such as the cosine similarity which seems to use a different model
>
> We used Spacy's pretrained pipeline to compute cosine similarity between two text segments. We mentioned this in Appendix B.
>
> > Reproducibility: The settings of parameters are underspecified or subjectively determined; the training/evaluation data are not widely available.
>
> We provided the full codebase and datasets used in the Supplementary along with the hyperparameter details in Appendix C.
>
> We commit to include the additional experimental results presented in the response in the revised version of our paper.

---

### Meta-Review · Area_Chair_FzWv · 2023-09-23

**Recommendation:** 4

**Metareview:**

The reviewers have a consensus that work is sound and exciting. The finetuning comparison as well as the clarrification in the review discussion should be included in further edits of the paper.

---

### Decision · Program_Chairs · 2023-10-07

**Decision:**

Accept-Main

**Comment:**

The reviewers have a consensus that work is sound and exciting. The finetuning comparison as well as the clarrification in the review discussion should be included in further edits of the paper.